# Distribution of nematophagous fungi and soil-transmitted helminths in outdoor built environments across Latin America

Rojelio Mejia[1]*, Eva Mereles Aranda[2], Leticia Ojeda[2], Sandra Ocampos Benedetti[2], Janitzio J. Guzman[3], Barton Slatko[1], Cristina Almazan[4], Melisa Diaz-Fernandez[4], Ruben Cimino[4], Marisa Juarez[4], Natalia Montellano Duran[5], Estefania Lorena Mansilla Flores[6], Paola Andrea Vargas[5], Amandeep Kaur[1], Nestor L. Uzcategui[1], Lucia Estela Mejia[1], Katherine Elizabeth Keegan[1], Emilio Rey Mejia[7], Chiara Cássia Oliveira Amorim[8], Stefan M. Geiger[8], Ricardo T. Fujiwara[8], Luz Marina Llangarí-Arizo[9,10], Andrea Lopez[10], Natalia Romero-Sandoval[10,11], Irene Guadalupe[12], Liliana E. Villanueva-Lizama[13], Julio Vladimir Cruz-Chan[13], Maritza Dalí Camones Rivera[14], Eddyson Montalvo Sabino[15], Carlos Pineda[14], Eric J. Wetzel[16], Philip J. Cooper[10,17]

1 Department of Pediatrics – Tropical Medicine, Baylor College of Medicine, Houston, Texas, United States of America, 2 Centro de Investigaciones Medicas, Facultad de Ciencias de la Salud, Universidad Nacional Del Este, Paraguay, 3 Department of Medicine – Infectious Diseases, Baylor College of Medicine, Houston, Texas, United States of America, 4 Universidad Nacional de Salta, Facultad Regional Orán, Instituto de Investigaciones de Enfermedades Tropicales (IIET-CONICET), Orán, Salta, Argentina, 5 Centro de Investigación en Biotecnología (BIOCEN), Universidad Católica Boliviana San Pablo, Santa Cruz de la Sierra, Bolivia, 6 Ingeniería en Biotecnología, Universidad Católica Boliviana San Pablo, Santa Cruz de la Sierra, Bolivia, 7 Robert Turner College and Career High School, Pearland, Texas, United States of America, 8 Departamento de Parasitologia, Instituto de Ciências Biológicas, Universidade Federal de Minas Gerais, Belo Horizonte, Brazil, 9 Escuela de Medicina Veterinaria, Universidad Internacional del Ecuador UIDE, Ecuador, 10 Escuela de Medicina, Universidad Internacional del Ecuador UIDE, Ecuador, 11 Grups de Recerca d'America i Africa Llatines-GRAAL, 12 IESS Hospital, Puyo, Pastaza Province, Ecuador, 13 Laboratorio de Parasitología, Centro de Investigaciones Regionales "Dr. Hideyo Noguchi", Universidad Autónoma de Yucatán, Mérida, México, 14 Facultad de Medicina Veterinaria y Zootecnia, Universidad Nacional Hermilio Valdizán, Huánuco, Peru, 15 Instituto de Investigación en Enfermedades Tropicales, Universidad Nacional Toribio Rodríguez de Mendoza, Amazonas, Perú, 16 Department of Biology, and Global Health Initiative, Wabash College, Crawfordsville, Indiana, United States of America, 17 School of Health and Medical Sciences, City St George's University of London, London, United Kingdom

* rojelio.mejia@bcm.edu

## Abstract

### Background

Soil-transmitted helminths (STHs) are among the most common global parasitic infections, represent a significant worldwide public health burden, and remain a source of considerable morbidity in Latin America. Nematophagous fungi (NF), such as *Arthrobotrys oligospora,* naturally inhabit many soil types and are known for their ability to trap and kill nematodes using specialized hyphal structures or secreted enzymes and metabolites. As they prey on different developmental stages

**Data availability statement:** All data are in the manuscript and the Supporting information files.

**Funding:** The author(s) received no specific funding for this work.

**Competing interests:** The authors have declared that no competing interests exist.

of helminths in soil, they may represent an ecological factor influencing helminth persistence and transmission dynamics.

## Methods

Using an *in vitro* test, *Toxocara cati* eggs were exposed to *A. oligospora*. By using a flotation, filtration, and bead-beating disruption technique, parasite and fungal DNA were collected and detected by multi-parallel real-time quantitative PCR (qPCR). Similar methods were used to extract DNA from soil samples outside built environments across seven Latin American countries, including Argentina, Bolivia, Brazil, Ecuador, Mexico, Paraguay, and Peru.

## Results

*In vitro* testing showed a 40.1% reduction in viable eggs *in* the presence of *A. oligospora,* as determined by qPCR (P = 0.0212). Comparing the impact of *A. oligospora* on *T. cati* over 14 days revealed a decrease in *T. cati* DNA concentration compared to control groups (P = 0.0039)*.* Using qPCR to detect *A. oligospora,* there was a 62.4% decrease in the mean *A. oligospora* DNA at 14 days. The co-occurrence of NF and STH was evaluated in 805 soil samples from seven Latin American countries representing distinct geoclimatic settings. We observed a significant reduction in helminth abundance (P < 0.05), including *Ascaris, Strongyloides, Toxocara,* and any helminth.

## Conclusion/significance

The ubiquitous presence of *A. oligosp*ora in soils and inverse association with STH parasite detection suggest a potential role in environmental helminth transmission patterns.

### Author summary

Soil-transmitted helminths (STH) remain a major cause of illness in many regions because their eggs and larvae remain viable in soil, continually driving reinfection. Current human treatment alone cannot interrupt this cycle. We evaluated *Arthrobotrys oligospora*, a nematophagous fungus naturally present in soil, to better understand its interactions with STH in environmental settings. *In vitro*, the fungus reduced helminth DNA concentrations, and its presence was detected in soils from seven Latin American countries. The fungus grows toward and penetrates parasites, including *Ascaris, Strongyloides*, and *Toxocara*. Its natural presence in soils in outdoor built environments was associated with decreased STH detection. These findings suggest that soil-dwelling nematophagous fungi may represent an ecological factor influencing environmental helminth persistence and transmission dynamics.

## Introduction

As major human pathogens worldwide, soil-transmitted helminths (STH) are estimated to affect over 1.5 billion people, mainly in populations living in poverty and with inadequate access to healthcare [1,2]. STH parasites include an adult intestinal stage, with viable eggs released into the environment upon host defecation. In areas of poor sanitation, these eggs reside in the soil and embryonate after several weeks. Individual infection can occur through two main routes: most commonly, by ingestion of eggs through play (usually by children), or by other direct and indirect oral contact with contaminated soil. In the case of hookworms and *Strongyloides*, larvae can also hatch in the soil and penetrate the skin, often after walking on contaminated soil. Without treatments, the infections can lead to severe morbidity, including colitis, anemia, malnutrition, and, in children, delays in growth and cognitive development [3,4].

It has become clear that traditional approaches addressing human and animal infection, focusing on administering anti-parasitic treatments and chemical treatments of soils, are inadequate, due to several challenges: insufficient coverage and compliance with treatments; development of drug resistance [5]; reinfection due to persistent environmental reservoirs of infective eggs; anti-parasitic drugs do not affect eggs or larvae in soil. An alternative perspective is to examine natural biological interactions that influence the persistence of parasite stages within soil environments [6].

The association for STH intervention with Nematophagous fungi (NF) should consider toxocariasis, as it poses a significant hazard to human and animal health, causing severe morbidity, including eosinophilia, asthma, colitis, visual or neuropathology due to larval migrations, malnutrition, iron deficiency, stunting, and, in children, delays in physical and cognitive development [7–9]. As the life cycle persists through reinfection via contaminated soil, repeated treatments with anti-parasitic drugs designed to kill adult worms are required. Throughout Latin America, *Toxocara* prevalence in outdoor built areas is high, with reports of 50% to 100% in public parks [10].

One such unique association employs NF species that have evolved to use nematodes in the environment as a food source [11]. NF isolated from agricultural fields, forests, and compost soils has a history of use as agents for reducing plant and veterinary animal parasitic nematodes [11–18]. However, their ecological role in STH transmission dynamics has not been systematically explored. As native soil organisms that act independently of host treatment or behavior, NF are well-suited for studying environmental influences on parasite persistence.

One common NF species is *Arthrobotrys oligospora,* one of the first NFs to be recognized [11,19]. While *A. oligospora* primarily uses nematode trap formation to capture and inactivate nematode eggs, along with other NF species, it uses at least one of 4 main mechanisms: nematode trapping using hyphae, spore-based endoparasitic activity, invasion of eggs or larvae, or toxin secretion before invasion. Much is known about the diversity, taxonomy, biology, and ecology of NF, and the molecular/biochemical mechanisms of NF are under active investigation [20,21]. For this pilot project only *A. oligospora* was evaluated.

We aimed first to confirm previous experiments to evaluate and quantify the effect of NF on the commonly found STH, *Toxocara cati,* and, secondly, to confirm and extend these observation*s* by assaying NF activity in soil samples from seven Latin American countries. We utilized the commonly found NF species, *Arthrobotrys oligospora* [22]. The results of this study highlight the potential importance of ecological processes in shaping STH transmission dynamics in soil. These findings support a broader understanding of how naturally occurring environmental organisms may influence helminth persistence outside the host and underscore the value of incorporating environmental biological interactions into models of parasite transmission.

## Methods

### Ethics approval and consent to participate

No human subjects were enrolled in this environmental study, and no Ethics Committee approval was required. Residents of the homes were invited to join the study and granted permission to collect soil samples.

## Biological samples

*Arthrobotrys oligospora* obtained from ATCC (American Type Culture Collection, Manassas, VA, USA; strain 24927). Primer-specific qPCR verified that *A. oligospora* had been grown on yeast peptone dextrose (YPD) plates before use. *A. oligospora* inoculum was transferred and cultured using sterile technique, in a liquid buffered glycerol-complex medium (BMGY), inoculated and grown at 25°C. Helminth eggs include *Toxocara cati* that were collected from infected cats' feces. *Toxocara*-infected feces were collected from infected cats participating in ongoing research studies and stored at 4°C. Animal feces are weighed and suspended in Feca-Med (Vedco, Inc., St. Joseph, MO) containing Sodium Nitrate (specific gravity 1.3). After agitation and centrifugation, the liquid layer was filtered to retain *Toxocara* eggs. These were counted by Counting Chamber (Advanced Equine Products, Issaquah, WA) and used for soil studies.

## Latin America multi-country sampling of built environments

A total of 805 soil samples from 218 external built environments in seven Latin American countries were sampled for STH, *Toxocara* species, and *A. oligospora.* Countries included are Argentina, Bolivia, Brazil, Ecuador, Mexico [23], Paraguay, and Peru [24] (Table 1). Samples were collected from 1 cm deep surface scrapings, and soil was stored at 4°C before processing. Built-environment locations were selected from prior community health campaigns in resource-limited settings. All dirt samples were collected outside participants' houses, including the entrance, latrine, and patio, except for 45 samples from seven city parks in Huánuco, Peru [24]. These locations are in resource-poor areas throughout Latin America, with outdoor latrines and limited access to proper sanitation. Latrines were limited-service latrines on the Joint Monitoring

**Table 1. Environmental Sampling and the number of samples for each Latin American country. Ecosystem map in S1 Fig.**

| Country | Date of collection | Number of Samples (805) | Number of Built Environments (218) | Ecosystem | Climate Region (Köppen-Geiger code) |
|---|---|---|---|---|---|
| Argentina | November 2023 | 111 | 28 | Tropical Savanna | Temperate Dry winter Hot summer (Cwa) |
| Bolivia | June 2024 | 93 | 25 | Tropical Dry Forest | Tropical Savanna (Aw) |
| Brazil | July 2023 | 88 | 40 | Savanna Woodlands | Tropical Savanna (Aw) |
| Ecuador | August 2023, July 2024 | 101 | 21 | Warm Temperate Moist Forest | Tropical Rainforest (Af) |
| Mexico | August 2018 | 77 | 34 | Tropical Dry Forest | Tropical Savana (Aw) |
| Paraguay | June 2024, 2025 | 137 | 27 | Subtropical Forest | Humid Subtropical (Cfa) |
| Peru | August 2023 | 198 | 43 | Mountainous highlands | Temperate Highland (Cwb) |
| | | | | Amazon rainforest | Tropical Rainforest (Af) |

Programme (JMP) sanitation ladder, about 2 – 5 meters from the house's entrance. Permission was obtained from tenants and local Public Health officials.

## DNA isolations

The DNA concentration technique uses parasite flotation and filtration to concentrate parasite DNA from experimental and control soil samples before DNA extraction, as previously described [24]. All DNA extraction was performed at each field site and varied as reported. Briefly, Phosphate-buffered saline (Alfa Asesar, Ward Hill, MA) with 0.05% TWEEN (Sigma-Aldrich, St. Louis, MO) was added to a 50 mL tube containing approximately 25 g of soil sample. Samples were weighed and recorded. The samples were vortexed for 5 minutes using a Tornado II portable paint shaker with a 115-volt motor (Blair Equipment Company, Swartz Creek, MI), then centrifuged at 500 $g$ for 5 minutes, and the supernatant containing the debris was discarded. To float helminth eggs and larvae, 10 mL of a 35.6% $NaNO_3$ solution (Vedco, St. Joseph, MO) (USA, Ecuador, and Mexico) or 530 mg/mL of sugar (Argentina, Bolivia, Brazil, Paraguay, and Peru), with a specific gravity of 1.3 measured with a hydrometer (SP Scienceware, Wayne, NJ) and added to the pellet in each conical centrifuge tube. The solution was vortexed in a portable shaker for 5 minutes, then centrifuged for 5 minutes at 500 $g$. The supernatant for each sample containing the floated parasites was transferred to a filtration apparatus. The filtration apparatus consists of a 20-mL syringe fitted with a 3 µm pore nitrocellulose filter (Millipore Sigma, Burlington, MA), which is small enough to retain all tested parasites. The filtration apparatus was attached to a vacuum manifold, which in turn was connected to a two-stage rotary vane vacuum pump (ELITech, Puteaux, France). Filtration with a vacuum pressure as low as 25 µm Hg was performed until the eluent had passed through the filter. The MP Fast SpinKit for Soil (MP Biomedicals, Santa Ana, CA) was used to extract DNA from parasites retained on a nitrocellulose filter [24]. An internal control DNA was used as an exogenous control to confirm efficient extraction [25]. Heat disruption at 90°C for 10 minutes in a dry bath incubator was followed by mechanical disruption using the MP FastPrep 24-5G disruptor (MP Biomedicals, Santa Ana, CA) at speed 6 for 40 seconds (USA, Ecuador) or a Disruptor Genie (Scientific Industries, Bohemia, NY) at 3,000 rpm for 5 minutes (Argentina, Bolivia, Brazil, Mexico, Paraguay, and Peru). All DNA eluent was stored at -20°C; information is listed in the Environmental Microbiology Minimum Information (EMMI) (S1 Table). The eluent volume was measured and collected from all seven Latin American countries, spotted onto 0.2 µm filter paper (Millipore, Merck KGaA, Darmstadt, Germany), air-dried, and shipped at ambient temperature to Baylor College of Medicine, Houston, TX, USA. Once received, DNA was extracted from the filter papers by overnight room-temperature elution using the same volume of elution buffer (MP Biomedicals).

## Multi-parallel real-time quantitative PCR (qPCR) and primers

DNA collected was analyzed using a multi-parallel real-time quantitative PCR. For each reaction, a 7-µL mixture consisted of 5 µL TaqMan Fast Advanced Master Mix (Applied Biosystems, Foster City, CA) with 900 nM each of forward and reverse primers (and FAM probe with a minor groove binder and nonfluorescent quencher (100 nM final concentration) [26]. 2 µL of extracted DNA was added to each reaction mixture. All reactions were run on a 96-well plate (except in Argentina), with a standard curve prepared using parasite plasmid or fungal genomic DNA. Each plate had a positive (plasmid) and a negative (no template) control. All DNA samples were tested with an internal control to confirm DNA presence [25]. All plates for fungus were processed on a QS7 Pro Real-Time PCR System (Thermo Fisher Scientific, Waltham, MA) in Houston, Texas, including helminths from Bolivia, Mexico, Paraguay, and Peru. In Ecuador and Brazil, the ABI 7500 Real-Time PCR System (Thermo Fisher Scientific) was used for parasites. In Argentina, the Chia Portable Real-time PCR (Chia Bio, Santa Clara, CA) was used in a 16-well format with two known parasite standard concentrations and two negative control wells. We used qPCR and microscopy to quantify the experimental results. Helminths tested include *Ancylostoma* species*, Ascaris lumbricoides, Necator americanus, Strongyloides stercoralis, Toxocara canis, Toxocara cati,* and *Trichuris trichiura* (S2 Table).

qPCR for *A. oligospora* quantification used primers as described [20].
Forward: CGG TTT GCT GTT GCA GCT TGT T
Reverse: GGT TCA CAA AGG GTT TAC CAG G
Probe: FAM -CTG TCT TCC GGT TGG TAA GC

### *In vitro* tests of *A. oligospora* on *T. cati* eggs

For *in vitro* experiments, commercial all-purpose soil (Garden Soil, Miracle-Gro, Marysville, OH) was spread on petri plates (approximately 25 g). *A. oligospora* was cultured in BMGY at 25°C with agitation until an optical density of 0.5 at 600nm, as measured by an Epoch plate reader (BioTek, Winooski, VT). Plates were then incubated at 25°C for 5 days before the addition of *T. cati* eggs. For all *in vitro* experiments, 1 mL of *A. oligospora* and 3,000 *T. cati* eggs were used per plate.

### Statistical analysis

One-way and two-way ANOVA were used for *in vitro* testing, along with Mann-Whitney tests and log10 transformations for two-way ANOVA. For the Latin American soil samples, odds ratios were calculated using either the Chi-square or Fisher's exact test. Any value $P < 0.05$ was considered significant. Calculations and graphing were performed using GraphPad Prism V.10.6.1 (San Diego, CA).

## Results

### *In vitro* experiments

*In vitro* experiments were first performed to confirm and quantify the activity of *A. oligospora*, using *T. cati* eggs. Microscopy visualized fungal predation on *Toxocara cati* eggs, and qPCR confirmed the complete reduction of *Toxocara* DNA on YPD plates. Control plates contained no fungus. There was no visible interaction between the fungus and *T. cati* eggs at the start of the experiment (Fig 1.1). At 4 hours, fungal hyphae were seen growing towards each *T. cati* egg within proximity of the fungus (Fig 1.2). After 2 days, fungal penetration was observed (Fig 1.3). The control plates had larval hatching at day 2 (Fig 1.4).

In a subsequent experiment, 45 plates containing 3,000 *T. cati* eggs each, with and without fungus, were cultured for 9 days, with a decrease of *T. cati* DNA (Fig 2A) in combination with *A. oligospora*. The control (no added fungus) group had a median DNA value of 103.1 fg/µL. In comparison, the experimental group had a median DNA value of 61.73 fg/µL, a decrease of 41.37 fg/µL (40.1% reduction) (P = 0.0212). A follow-up 14-day study on 63 separate and individual plates of 3,000 *T. cati* eggs each, with *A. oligospora* added to 33 plates. DNA was subsequently isolated from experimental and control plates (no fungus added) on days 0, 3, 7, 9, 11, and 14. Using two-way ANOVA with log10-transformed *T. cati* DNA concentration as the outcome, there was a significant effect of fungus on *T. cati* DNA (P = 0.0039), some evidence of a time trend (P = 0.0654), but no treatment-time interaction (P = 0.8928) (Fig 2B).

### Fungal testing in soil

The same *T. cati* plates (Fig 2B) were also tested for the presence and quantity of *A. oligospora* DNA. DNA was detected in all spiked experimental samples, with decreasing concentrations of *A. oligospora* DNA over time, indicating transient soil colonization, likely due to the natural life span of fungi in soil. There was a 62.4% decrease in the mean *A. oligospora* DNA at 14 days, although using a log10-transformed two-way ANOVA was not significant for *A. oligospora* over time (P = 0.3422) (Fig 3). Even with a reduction in fungal DNA, there was a consistent decrease in *T. cati* DNA compared to the paired control group (Fig 2B). Several control samples also naturally contained *A. oligospora* DNA present in the added soil. As a proof-of-principle test, control samples that contained *A. oligospora* DNA were removed from the *in vitro*

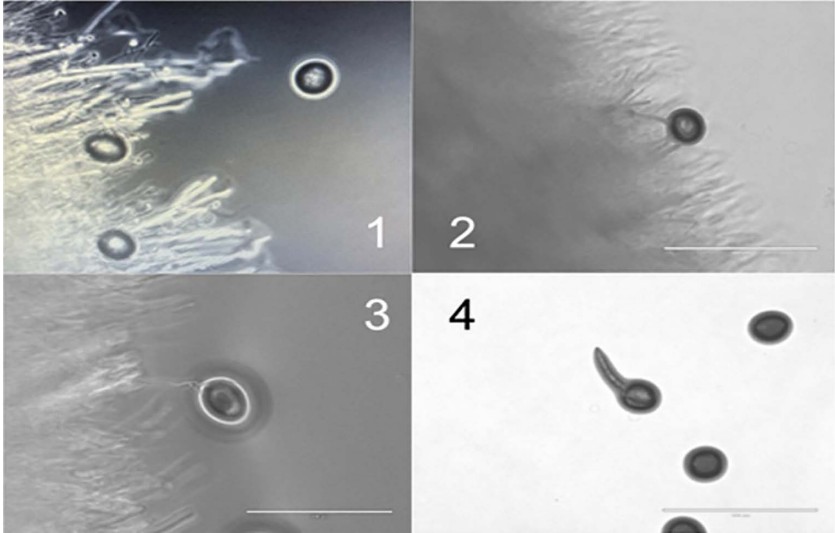

**Fig 1. *A. oligospora* grows towards and invades *T. catti*.** 1) 7 days of fungal growth and initial addition of *T. cati* show no interaction. 2) After 4 hours, the fungus grows towards *T. cati* eggs. 3) At 2 days, fungal hyphae penetrate *T. cati* eggs. 4) A control shows larva and egg hatching of living *T. cati* eggs.

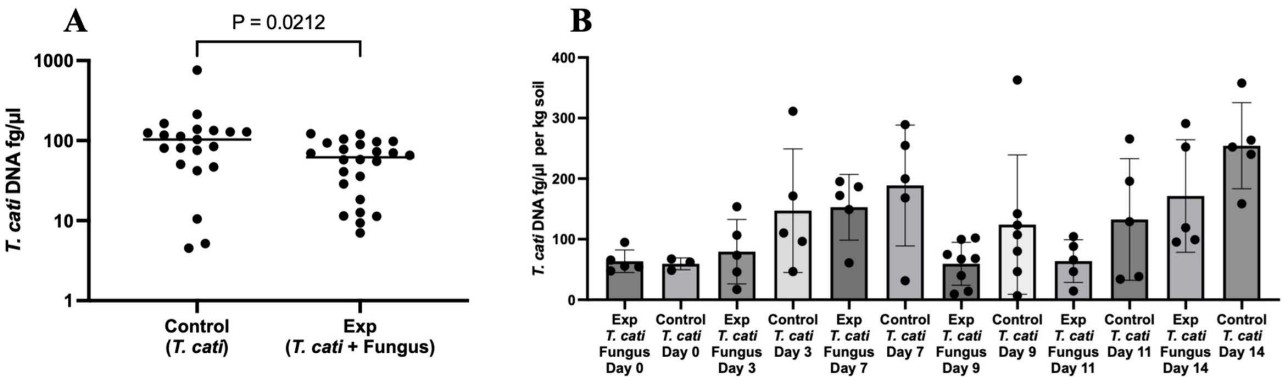

**Fig 2. A.** The presence of *A. oligospora* significantly reduced *T. cati* DNA levels. **B.** Besides, at day 0, *A. oligospora* significantly decreased the concentration of *T. cati* (Exp = Experiment).

analysis. The two-way ANOVA with log 10-transformed *T. cati* DNA remained significant on the impact of fungus on *T. cati* DNA (P = 0.0148) and no significance on time (P = 0.5318) or interaction (P = 0.3096) (S2 Fig).

## Soil testing from seven Latin American Countries

Soil sampling for STH and *A. oligospora* was conducted at sites in seven Latin American countries (Table 1) (S1 Fig). The results on contamination rates per sample and per built environment, including the mean and range of organism DNA concentrations, showed a wide range across countries (Table 2).

The overall occurrence of parasites and *A. oligospora* per kg soil across all seven countries showed a large range between helminths and *A. oligospora* (Fig 4). Individual country data is presented in (S3 Fig). No comparisons between

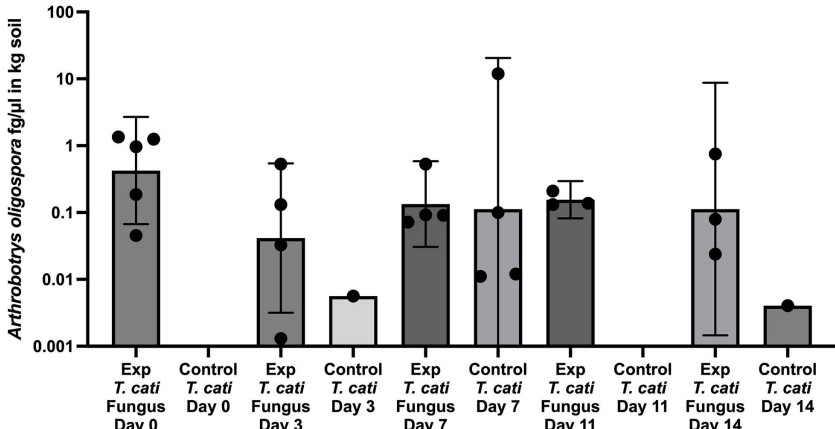

**Fig 3. *A. oligospora* DNA can be detected in all spiked experimental samples with decreasing concentration of DNA over time.** There is a 62.4% decrease in the mean DNA at 14 days (P = 0.3422). Several control samples naturally contained *A. oligospora* DNA (Exp = experiment).

parasite DNA (fg/µl per kg soil) are possible, since the qPCR DNA sequence targets are from different regions on the helminth genomes and cannot be adequately compared across organisms.

Odds ratios and confidence intervals estimate the strength of the association between qualitative detection of *A. oligospora* DNA and that of individual STH parasites. *A. oligospora* was present in 7.4% (5.4 to 11.7%) of samples in all countries combined. When present, the odds ratio analysis shows reduced detection of *Ascaris, Strongyloides, and Toxocara* species. *In toto*, pooled data show a reduction across all helminths (Table 2). Combining all seven country data, *A. oligospora* was protective against *Ascaris* (OR 0.24, 95% CI 0.11–0.51, P < 0.0001), *Strongyloides* (OR 0.41, 95% CI 0.18–1.0, P = 0.03), *T. cati* (OR 0.21, 95% CI 0.061–0.74, P = 0.042), and *Toxocara* species (OR 0.37, 95% CI 0.16-0.93, P = 0.02) A composite "any helminth" outcome also showed significant protection (OR 0.41, 95% CI 0.24–0.71, P = 0.0015). Overall, the presence of *Arthrobotrys oligospora* was associated with a decrease in the detection of STH parasites (Table 3) (Fig 5).

## Discussion

Nematophagous fungi have been suggested as biologically active organisms affecting plant and animal parasitic nematodes [27]. This exploratory study, which included *in vitro* analysis in a controlled laboratory setting complemented by field observations from seven Latin American locations, showed an inverse association between *A. oligospora* and STH – the presence of *A. oligospora* was consistently associated with lower helminth detection rates. Significant associations with *Ascaris*, *Strongyloides,* and *Toxocara* suggest that NF may represent an ecologically relevant factor influencing environmental helminth persistence and transmission.

This study represents a wide variety of ecosystems and climate regions across seven Latin American countries that are endemic for parasitic infections in humans and animals [3,28–33] (Table 1, S1 Fig). The life cycles of these STHs depend on soil type, environment, and climate. These include temperature, humidity, soil moisture, and outdoor temperatures. Also, the lack of proper sanitation in these endemic regions, including the use of outdoor latrines or open-air defecation, likely augments the environmental reservoir of STH and increases the risk of transmission. In 2020, 23.1% of people worldwide had limited access to proper sanitation, indicating that almost 1 in 4 people are at risk of STH infections [34].

The original studies on NFs focused on helminth larvae and described a fungal lattice network that would entrap them [35]. Although recent studies also explore the penetration and digestion of helminth eggs [19]. Interestingly, a series of *A. oligospora* G protein-coupled receptors has been described that may sense helminth pheromones, thereby directing

**Table 2. Prevalence of parasites and DNA concentrations in the soil.**

| Parasite/ fungus Country | Contamination Rate (Samples) | Contamination Rates (Built Environment) | DNA concentration in kg of soil (fg/µl), mean (range) |
|---|---|---|---|
| ***Ancylostoma species*** | | | |
| Overall | 2.6% (21/805) | 8.7% (19/218) | 368.8 (0.013 to 7479.9) |
| Argentina | 0% (0/111) | 0% (0/28) | 0 |
| Bolivia | 3.2% (3/93) | 12% (3/25) | 8.2 (0.042 to 24.6) |
| Brazil | 2.3% (2/88) | 5% (2/40) | 0.014 (0.013 to 0.015) |
| Ecuador | 7.9% (8/101) | 33.3% (7/21) | 23.4 (0.63 to 83.7) |
| Mexico | 1.3% (1/77) | 2.9% (1/34) | 7479.9 |
| Paraguay | 0% (0/137) | 0% (0/27) | 0 |
| Peru | 3.5% (7/198) | 13.9% (6/43) | 7.6 (0.25 to 36.7) |
| ***Ascaris lumbricoides*** | | | |
| Overall | 6.2% (50/805) | 13.8% (30/218) | 515.3 (0.026 to 9500.8) |
| Argentina | 9.0% (10/111) | 14.3% (4/28) | 203.9 (0.1 to 1494.9) |
| Bolivia | 1.1% (1/93) | 4% (1/25) | 1.2 |
| Brazil | 10.2% (9/88) | 17.5% (7/40) | 29.9 (0.03 to 274.9) |
| Ecuador | 9.9% (10/101) | 33.3% (7/21) | 35.6 (0.5 to 124.9) |
| Mexico | 9.1% (7/77) | 11.8% (4/34) | 3379.2 (336.1 to 9500.8) |
| Paraguay | 0% (0/137) | 0% (0/27) | 0 |
| Peru | 6.6% (13/198) | 16.3% (7/43) | 18.6 (0.16 to 152.7) |
| ***Necator americanus*** | | | |
| Overall | 2.2% (11/805) | 3.7% (8/218) | 0.47 (0.063 to 0.88) |
| Argentina | 0% (0/111) | 0% (0/28) | 0 |
| Bolivia | 0% (0/93) | 0% (0/25) | 0 |
| Brazil | 0% (0/88) | 0% (0/40) | 0 |
| Ecuador | 6.9% (7/101) | 19.1% (4/21) | 0.58 (0.26 to 0.88) |
| Mexico | 0% (0/77) | 0% (0/34) | 0 |
| Paraguay | 0.7% (1/137) | 3.7% (1/27) | 0.73 |
| Peru | 1.6% (3/198) | 7.0% (3/43) | 0.12 (0.063 to 0.23) |
| ***Strongyloides stercoralis*** | | | |
| Overall | 5.7% (46/805) | 15.1% (33/218) | 7568.1 (0.016 to 347895) |
| Argentina | 3.6% (4/111) | 7.1% (2/28) | 1.9 (0.02 to 7.4) |
| Bolivia | 7.5% (7/93) | 24% (6/25) | 0.57 (0.22 to 1.2) |
| Brazil | 14.8% (13/88) | 25% (10/40) | 18.0 (0.09 to 161.4) |
| Ecuador | 8.9% (9/101) | 33.3% (7/21) | 38656.1 (0.02 to 347895) |
| Mexico | 0% (0/77) | 0% (0/34) | 0 |
| Paraguay | 0.73% (1/137) | 3.7% (1/27) | 0.95 |
| Peru | 6.1% (12/198) | 16.3% (7/43) | 0.74 (0.016 to 3.5) |
| ***Toxocara canis*** | | | |
| Overall | 4.1% (33/805) | 11.5% (25/218) | 70888.8 (0.38 to 2188280) |
| Argentina | 3.6% (4/111) | 14.3% (4/28) | 470591 (300.8 to 2188280) |
| Bolivia | 4.3% (4/93) | 16% (4/25) | 181.8 (2.6 to 427.1) |
| Brazil | 0% (0/88) | 0% (0/40) | 0 |
| Ecuador | 11.9% (12/101) | 38.1% (8/21) | 4305.2 (18.3 to 39351.6) |

*(Continued)*

| Parasite/ fungus Country | Contamination Rate (Samples) | Contamination Rates (Built Environment) | DNA concentration in kg of soil (fg/µl), mean (range) |
|---|---|---|---|
| Mexico | 9.1% (7/77) | 8.8% (3/34) | 376.2 (1.6 to 1042.4) |
| Paraguay | 2.9% (4/137) | 14.8% (4/27) | 493.1 (5.7 to 1,054.1) |
| Peru | 1.0% (2/198) | 4.6% (2/43) | 133.8 (0.38 to 267.2) |
| *Toxocara cati* | | | |
| Overall | 13.7% (11/805) | 3.7% (8/218) | 6635.9 (918.7 to 62236) |
| Argentina | 0% (0/111) | 0% (0/28) | 0 |
| Bolivia | 1.1% (1/93) | 4% (1/25) | 62236 |
| Brazil | 0% (0/88) | 0% (0/40) | 0 |
| Ecuador | 9.9% (10/101) | 33.3% (7/21) | 1075.9 (918.7 to 1263.0) |
| Mexico | 0% (0/77) | 0% (0/34) | 0 |
| Paraguay | 0% (0/137) | 0% (0/27) | 0 |
| Peru | 0% (0/198) | 0% (0/43) | 0 |
| *Trichuris trichiura* | | | |
| Overall | 3.1% (25/805) | 10.6% 23/218) | 9571.6 (0.017 to 141398) |
| Argentina | 4.5% (5/111) | 14.3% (4/28) | 65.1 (25.9 to 130.2) |
| Bolivia | 5.4% (5/93) | 25% (5/25) | 19467 (0.11 to 97314) |
| Brazil | 0% (0/88) | 0% (0/40) | 0 |
| Ecuador | 4.0% (4/101) | 19.1% (4/21) | 35367.1 (1.05 to 141398) |
| Mexico | 2.6% (2/77) | 2.9% (1/34) | 0.68 (0.49 to 0.88) |
| Paraguay | 1.5% (2/137) | 7.4% (2/27) | 3.6 (0.57 to 6.6) |
| Peru | 3.5% (7/198) | 16.3% (7/43) | 21.2 to (0.016 to 104.1) |
| **Any helminth** | | | |
| Overall | 19.0% (153/805) | 42.2% (92/218) | |
| Argentina | 16.2% (18/111) | 35.7% (10/28) | |
| Bolivia | 21.5% (20/93) | 56% (14/25) | |
| Brazil | 23.9% (21/88) | 37.5% (15/40) | |
| Ecuador | 34.6% (35/101) | 81.0% (17/21) | |
| Mexico | 18.2% (14/77) | 17.6% (6/34) | |
| Paraguay | 5.8% (8/137) | 25.9% (7/27) | |
| Peru | 18.7% (37/198) | 53.5% (23/43) | |
| *Arthrobotrys oligospora* | | | |
| Overall | 7.4% (60/805) | 24.8% (54/218) | 5.55 (0.005 to 62.6) |
| Argentina | 11.7% (13/111) | 42.8% (12/28) | 0.99 (0.04 to 2.98) |
| Bolivia | 5.4% (5/93) | 16% (4/25) | 2.9 (0.69 to 4.72) |
| Brazil | 9.1% (8/88) | 20% (8/40) | 12.6 (1.6 to 62.7) |
| Ecuador | 7.9% (8/101) | 33.3% (7/21) | 18.9 (2.8 to 60.1) |
| Mexico | 6.5% (5/77) | 14.7% (5/34) | 0.043 (0.005 to 0.11) |
| Paraguay | 6.6% (9/137) | 29.6% (8/27) | 0.32 (0.056 to 1.7) |
| Peru | 6.1% (12/198) | 23.3% (10/43) | 4.2 (1.19 to 9.27) |

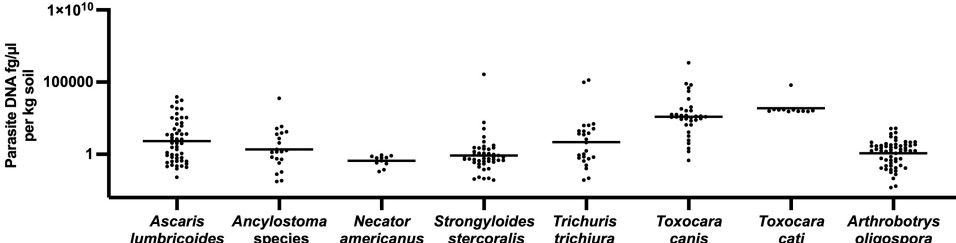

Argentina, Bolivia, Brazil, Ecuador, Mexico, Paraguay, Peru

**Fig 4. The combined helminths and *A. oligospora* DNA concentrations across all seven countries.**

**Table 3. The odds of detecting *A. oligospora* DNA but not helminth DNA across all seven Latin American countries combined. Chi-square analysis was used for *Ascaris, Strongyloides, Toxocara* species, and any helminth. Fisher's exact test was used for *Ancylostoma, Necator, Toxocara canis/cati*, and *Trichuris*.**

| Helminth | ODD RATIO | 95% CI Lower | 95% CI Upper | P value |
|---|---|---|---|---|
| *Ascaris lumbricoides* | 0.24 | 0.11 | 0.51 | <0.0001 * |
| *Ancylostoma* species | 0.66 | 0.19 | 3.37 | 0.66 |
| *Necator americanus* | 0.35 | 0.09 | 1.67 | 0.19 |
| *Strongyloides stercoralis* | 0.41 | 0.18 | 1.00 | 0.03 * |
| *Toxocara canis* | 0.43 | 0.16 | 1.06 | 0.091 |
| *Toxocara cati* | 0.21 | 0.061 | 0.74 | 0.042 * |
| *Trichuris trichiura* | 0.92 | 0.24 | 4.1 | 0.71 |
| *Toxocara* species | 0.37 | 0.16 | 0.93 | 0.020 * |
| Any helminth | 0.41 | 0.24 | 0.71 | 0.0015 * |

fungal growth towards the helminths [36]. These findings can be applied to our *in vitro* studies, in which *A. oligospora* grows towards and penetrates *T. cati* (Fig 1**).**

Our study describes the impact of *A. oligospora* on several helminths. It shows by association that both fecal-oral and dermal penetration of helminths can be reduced in the presence of NF. By association, these findings were consistent in all seven Latin American countries. In Ecuador, *Ascaris* levels were comparable between *A. oligospora*–positive and –negative samples (Fig 5). The observed statistical significance was driven primarily by the disproportionate number of *Ascaris*-negative samples in the *A. oligospora*–absent group (88) compared with the present group [3].

Interestingly, the helminths (*Necator* and *Strongyloides*) were found in the *A. oligospora* positive group in Paraguay, likely due to the low prevalence of helminths in Paraguay's specific ecosystem, and not found in the *A. oligospora* negative group (Fig 5).

The limitations of our study include a relatively small sample size within individual countries, which can skew comparisons; however, when combined, we did detect a significant impact of NF on helminths. Another limitation is that we are detecting parasite and NF DNA that can come from dead organisms and do not represent active killing of helminths. However, all soil samples were floated and then filtered, which should have allowed only intact eggs/larvae to survive, reducing the quantity of dead helminths for DNA extraction. Another limitation is how the samples were processed, which may decrease the helminth DNA detected by qPCR, since the NF is known to encage helminth eggs/larvae and could remove STH during extraction. However, significant mixing and soil washing should break up any NF-induced entrapment of helminths. There is also potential cross-reactivity with other helminth non-human pathogens. Many helminth species

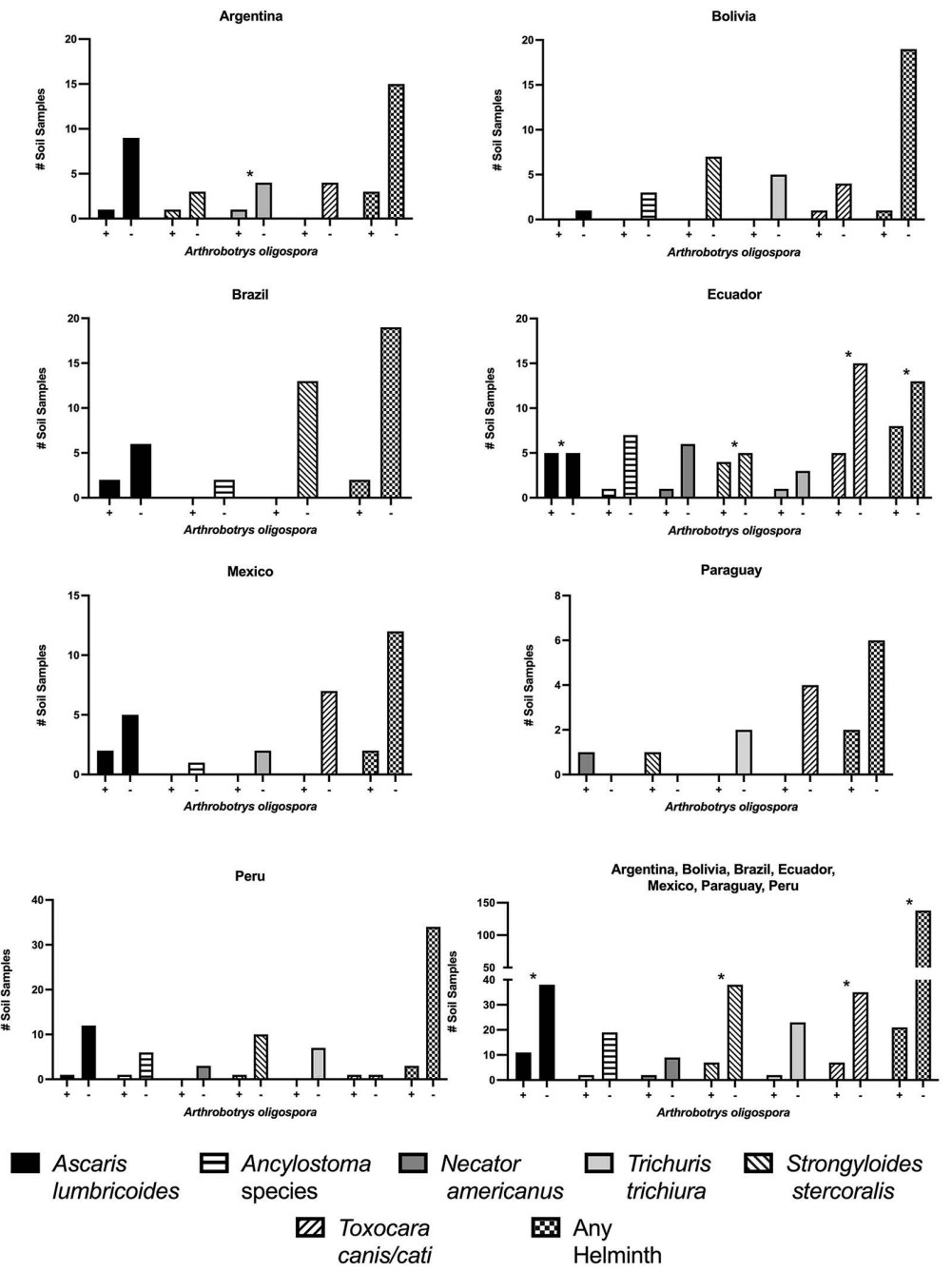

**Fig 5. The presence of *A. oligospora* DNA in soil samples was associated with a decrease in helminths throughout all seven countries.** In comparison to the lack of *A. oligospora* DNA in increased helminth numbers. The only significant differences were in Argentina (*Trichuris trichiura*), Ecuador (*Strongyloides stercoralis*, *Toxocara* species, and any helminth).

share nearly identical DNA sequences, even in the target regions of our primer/probe sets (S2 Table). Species such as *Ascaris ovis, Strongyloides ratti, Trichuris vulpis,* and others can also contaminate outdoor built environments accessible to feral animals. While cross-reactivity can occur, our study takes a one-health approach to the NF's impact on decreasing helminth infections in humans and animals. In this exploratory study, we did not measure or control for relevant

environmental factors, such as soil characteristics and climatic conditions, which are likely to affect STH detection across different sampling sites. Future studies will need to measure and control for such factors.

Soil-transmitted helminths persist in many regions due to the inherent biology of soil-based reinfection, along with barriers related to healthcare access, treatment adherence, and diagnostic limitations [37]. In addition, poverty and inadequate sanitation in endemic settings contribute to continued environmental contamination and reinfection cycles [38]. As such, our results suggest that NF *A. oligospora* is an ecologically relevant soil organism associated with reduced helminth burdens in environmental samples. We also show that *Arthrobotrys oligospora* NF is a naturally occurring soil component present in household soils where STH are present and is inversely correlated with STH contamination across diverse geographic regions, suggesting it is frequently present when parasite burdens are reduced or absent.

## Conclusions

In this exploratory study, we detected NF in soil samples from seven Latin American locations and provided evidence of an inverse association between NF and STH DNA. Further research will be required to examine the effects of soil composition and climate variation on NF activity and investigate how interactions between NF and STH may affect transmission dynamics and human exposure pathways. Longitudinal studies are warranted to evaluate the persistence of NF in soils, spatial-temporal variability, and their role within broader environmental transmission systems.

## Supporting information

**S1 Fig. Ecosystems and regions for seven Latin American countries.** Details are in Table 1 (World Terrestrial Ecosystems retrieved December 8, 2025 using ArcGIS Online by Environmental Systems Research Institute, https://www.arcgis.com/apps/mapviewer/index.html?layers=926a206393ec40a590d8caf29ae9a93e).
(TIFF)

**S1 Table. Environmental Microbiology Minimum Information (EMMI) for qPCR.**
(DOCX)

**S2 Table. Helminth target regions, primer sequences, and probe sequences for helminths for DNA amplification.**
(DOCX)

**S2 Fig. *Toxocara* concentrations after removing the Control samples that contained *Arthrobotrys oligospora*.**
(TIFF)

**S3 Fig. Parasite DNA concentration per Kg of soil across seven Latin American countries.** Values in Table 2.
(TIFF)

**S3 Table. Odds ratios of helminths and *A. oligospora* in each Latin American country.**
(DOCX)

**S1 Data. Fig 2A data.**
(XLSX)

**S2 Data. Fig 2B data.**
(XLSX)

**S3 Data. Fig 3 data.**
(XLSX)

**S4 Data. Fig 4 data.**
(XLSX)

## Acknowledgments

We wish to thank the study participants from all Latin American countries for welcoming us into their homes.

## Author contributions

**Conceptualization:** Rojelio Mejia, Eva Mereles Aranda, Leticia Ojeda.

**Data curation:** Rojelio Mejia.

**Formal analysis:** Rojelio Mejia.

**Investigation:** Rojelio Mejia, Eva Mereles Aranda, Leticia Ojeda, Sandra Ocampos Benedetti, Janitzio J Guzman, Barton Slatko, Cristina Almazan, Melisa Diaz-Fernandez, Ruben Cimino, Marisa Juarez, Natalia Montellano Duran, Estefanía Lorena Mansilla Flores, Paola Andrea Vargas, Amandeep Kaur, Nestor L. Uzcategui, Lucia Estela Mejia, Katherine Elizabeth Keegan, Emilio Rey Mejia, Chiara Cássia Oliveira Amorim, Stefan M. Geiger, Ricardo T Fujiwara, Luz Marina Llangarí-Arizo, Andrea Lopez, Natalia Romero-Sandoval, Irene Guadalupe, Liliana E. Villanueva-Lizama, Julio Vladimir Cruz-Chan, Maritza Dalí Camones Rivera, Eddyson Montalvo Sabino, Carlos Pineda, Eric J. Wetzel, Philip J Cooper.

**Methodology:** Rojelio Mejia, Eva Mereles Aranda, Leticia Ojeda, Cristina Almazan, Melisa Diaz-Fernandez, Marisa Juarez, Estefanía Lorena Mansilla Flores, Paola Andrea Vargas, Amandeep Kaur, Lucia Estela Mejia, Katherine Elizabeth Keegan, Emilio Rey Mejia, Chiara Cássia Oliveira Amorim, Luz Marina Llangarí-Arizo, Andrea Lopez, Irene Guadalupe, Liliana E. Villanueva-Lizama, Eddyson Montalvo Sabino.

**Supervision:** Rojelio Mejia, Eva Mereles Aranda, Ruben Cimino, Natalia Montellano Duran, Nestor L. Uzcategui, Stefan M. Geiger, Ricardo T Fujiwara, Natalia Romero-Sandoval, Julio Vladimir Cruz-Chan, Carlos Pineda, Eric J. Wetzel, Philip J Cooper.

**Writing – original draft:** Rojelio Mejia, Eva Mereles Aranda, Janitzio J Guzman, Barton Slatko, Cristina Almazan, Ruben Cimino, Paola Andrea Vargas, Nestor L. Uzcategui, Chiara Cássia Oliveira Amorim, Julio Vladimir Cruz-Chan, Carlos Pineda, Philip J Cooper.

**Writing – review & editing:** Rojelio Mejia, Eva Mereles Aranda, Janitzio J Guzman, Barton Slatko, Cristina Almazan, Ruben Cimino, Paola Andrea Vargas, Nestor L. Uzcategui, Chiara Cássia Oliveira Amorim, Julio Vladimir Cruz-Chan, Carlos Pineda, Philip J Cooper.

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
