## [Decision Letter · Decision Letter 0]

20 Jan 2026

Environmental Nematophagous Fungal Control of Soil-Transmitted Helminths in Contaminated Soils Across Latin America.

Dear Dr. Mejia,

Thank you for submitting your manuscript to PLOS Neglected Tropical Diseases. After careful consideration, we feel that it has merit but does not fully meet PLOS Neglected Tropical Diseases's publication criteria as it currently stands. Therefore, we invite you to submit a revised version of the manuscript that addresses the points raised during the review process.

Please submit your revised manuscript within by Mar 21 2026 11:59PM. If you will need more time than this to complete your revisions, please reply to this message or contact the journal office at plosntds@plos.org. Please include the following items when submitting your revised manuscript:

We look forward to receiving your revised manuscript.

Kind regards,

Robert Adamu SHEY, Ph.D.

Guest Editor

Jong-Yil Chai

Section Editor

Shaden Kamhawi

co-Editor-in-Chief

Paul Brindley

co-Editor-in-Chief

**Journal Requirements:**

At this stage, the following Authors/Authors require contributions: Rojelio Mejia, Eva Mereles Aranda, Leticia Ojeda, Sandra Ocampos Benedetti, Janitzio J Guzman, Barton Slatko, Cristina Almazan, Melisa Diaz-Fernandez, Ruben Cimino, Marisa Juarez, Natalia Montellano Duran, Estefanía Lorena Mansilla Flores, Paola Andrea Vargas, Amandeep Kaur, Nestor L. Uzcategui, Lucia Estela Mejia, Katherine Keegan, Emilio Rey Mejia, Chiara Cássia Oliveira Amorim, Stefan M. Geiger, Ricardo T Fujiwara, Luz Marina Llangarí-Arizo, Andrea Lopez, Natalia Romero-Sandoval, Irene Guadalupe, Liliana E. Villanueva-Lizama, Julio Vladimir Cruz-Chan, Maritza Dalí Camones Rivera, Eddyson Montalvo Sabino, Carlos Pineda, Eric J. Wetzel, and Philip J Cooper. Please ensure that the full contributions of each author are acknowledged in the "Add/Edit/Remove Authors" section of our submission form.

- ® on page: 10.

Potential Copyright Issues:

i) Figure S1. Please (a) provide a direct link to the base layer of the map (i.e., the country or region border shape) and ensure this is also included in the figure legend; and (b) provide a link to the terms of use / license information for the base layer image or shapefile. We cannot publish proprietary or copyrighted maps (e.g. Google Maps, Mapquest) and the terms of use for your map base layer must be compatible with our CC BY 4.0 license.

6) We note that your Data Availability Statement is currently as follows: "Data is contained in the manuscript and supplemental files.". Please confirm at this time whether or not your submission contains all raw data required to replicate the results of your study. Authors must share the “minimal data set” for their submission. PLOS defines the minimal data set to consist of the data required to replicate all study findings reported in the article, as well as related metadata and methods (https://journals.plos.org/plosone/s/data-availability#loc-minimal-data-set-definition).

**Reviewers' Comments:**

Reviewer's Responses to Questions

**Key Review Criteria Required for Acceptance?**

**Methods**

-Are the objectives of the study clearly articulated with a clear testable hypothesis stated?

-Is the study design appropriate to address the stated objectives?

-Is the population clearly described and appropriate for the hypothesis being tested?

-Is the sample size sufficient to ensure adequate power to address the hypothesis being tested?

-Were correct statistical analysis used to support conclusions?

-Are there concerns about ethical or regulatory requirements being met?

Reviewer #1: (No Response)

Reviewer #2: (No Response)

Reviewer #3: This study clearly states its objectives. The methods section is generally clear, but I have a couple of questions:

Please check that reference 24 is correct.

Could you please provide more details about the sampling sites, i.e., whether they were public areas such as parks or streets; and if there were any specific conditions that facilitated STH contamination?

**Results**

-Does the analysis presented match the analysis plan?

-Are the results clearly and completely presented?

-Are the figures (Tables, Images) of sufficient quality for clarity?

Reviewer #1: (No Response)

Reviewer #2: (No Response)

Reviewer #3: The results are clearly presented

**Conclusions**

-Are the conclusions supported by the data presented?

-Are the limitations of analysis clearly described?

-Do the authors discuss how these data can be helpful to advance our understanding of the topic under study?

-Is public health relevance addressed?

Reviewer #1: (No Response)

Reviewer #2: (No Response)

Reviewer #3: Although the statements on lines 400-406 are correct, please support them with ad hoc references.

The conclusion is appropriate.

**Editorial and Data Presentation Modifications?**

Reviewer #1: (No Response)

Reviewer #2: (No Response)

Reviewer #3: This manuscript is interesting and presents new knowledge through a little-studied approach on the potential future use of Arthrobotrys oligospora for HTS control.

**Summary and General Comments**

Reviewer #1: This manuscript studies the potential role of nematophagous fungi as an environmentally sustainable approach to reduce soil-transmitted helminths (STHs). The combination of laboratory-based experiments and multi-country environmental sampling is ambitious and, in principle, could provide valuable insights into parasite control at the soil level. However, in its current form, the study suffers from substantial conceptual, methodological, and interpretational limitations that critically undermine the validity of its conclusions. Here below are my main concerns:

The manuscript combines controlled in vitro experiments with a large observational soil survey; however, these two components are not mechanistically or causally integrated. The in vitro assays demonstrate fungal–nematode interactions under highly artificial conditions, whereas the environmental study relies solely on cross-sectional DNA detection. No evidence is provided that the mechanisms observed in vitro operate under natural soil conditions, rendering the linkage between the two datasets speculative.

Another major methodological flaw exists in the in vitro soil-based experiments. Commercial garden soil was used, and several control samples were found to naturally contain A. oligospora DNA. As a result, the control condition is not fungus-free, and qPCR cannot distinguish background fungal DNA from experimentally introduced fungus. This fundamental confounding undermines the validity of the reported reduction in Toxocara cati DNA.

In the environmental study, the nematode qPCR assays are acknowledged to lack strict species specificity, yet the results are interpreted at the species level and extrapolated to human health relevance. In addition, only A. oligospora was assayed despite the known diversity of nematophagous fungi (e.g. Duddingtonia flagrans, Monacrosporium spp., Arthrobotrys spp.) in soil, making broad conclusions about fungal-mediated control unjustified.

Finally, the soil data are purely correlational, based on DNA detection that does not indicate organism viability or active killing, and no multivariable analyses were performed to account for major environmental confounders. Despite these limitations, the manuscript repeatedly implies active biological control of STHs in the environment, which is not supported by the presented data.

The central conclusions regarding environmental control of soil-transmitted helminths by Arthrobotrys oligospora substantially exceed what can be supported by the current data. Addressing these limitations would require extensive redesign of both laboratory and field components of the study. Therefore, I do not consider the manuscript suitable for publication in its present form.

Reviewer #2: (No Response)

Reviewer #3: This manuscript is well written and presents novel results. I suggest clarifying a few minor comments before publication.

PLOS authors have the option to publish the peer review history of their article (what does this mean? ). If published, this will include your full peer review and any attached files.

**Do you want your identity to be public for this peer review?** For information about this choice, including consent withdrawal, please see our Privacy Policy .

Reviewer #1: No

Reviewer #2: No

Reviewer #3: No

**Figure resubmission:**
---

## [Editor Report · Decision Letter 1]

2 Feb 2026

Dear Dr Mejia,

We are pleased to inform you that your manuscript 'Distribution of Nematophagous Fungi and Soil-Transmitted Helminths in Outdoor Built Environments Across Latin America' has been provisionally accepted for publication in PLOS Neglected Tropical Diseases.

Best regards,

Robert Adamu SHEY, Ph.D.

Guest Editor

Jong-Yil Chai

Section Editor

Shaden Kamhawi

co-Editor-in-Chief

Paul Brindley

co-Editor-in-Chief

---

## [Editor Report · Acceptance letter]

Dear Dr Mejia,

We are delighted to inform you that your manuscript, "

Distribution of Nematophagous Fungi and Soil-Transmitted Helminths in Outdoor Built Environments Across Latin America," has been formally accepted for publication in PLOS Neglected Tropical Diseases.

Best regards,

Shaden Kamhawi

co-Editor-in-Chief

Paul Brindley

co-Editor-in-Chief
